# Characteristics of the Salivary Microbiota in Periodontal Diseases and Potential Roles of Individual Bacterial Species To Predict the Severity of Periodontal Disease

Suk Ji,[a] Joong-Ki Kook,[b] Soon-Nang Park,[b] Yun Kyong Lim,[b] Geum Hee Choi,[a] Jae-Suk Jung[a]

[a]Department of Periodontology, Institute of Oral Health Science, Ajou University School of Medicine, Suwon, Republic of Korea
[b]Korean Collection for Oral Microbiology, Department of Oral Biochemistry, School of Dentistry, Chosun University, Gwangju, Republic of Korea

Suk Ji and Joong-Ki Kook contributed equally to this work. Author order was determined based on alphabetical order.

**ABSTRACT** The purposes of this study were to examine the compositional changes in the salivary microbiota according to the severity of periodontal disease and to verify whether the distribution of specific bacterial species in saliva can distinguish the severity of disease. Saliva samples were collected from 8 periodontally healthy controls, 16 patients with gingivitis, 19 patients with moderate periodontitis, and 29 patients with severe periodontitis. The V3 and V4 regions of the 16S rRNA gene in the samples were sequenced, and the levels of 9 bacterial species showing significant differences among the groups by sequencing analysis were identified using quantitative real-time PCR (qPCR). The predictive performance of each bacterial species in distinguishing the severity of disease was evaluated using a receiver operating characteristic curve. Twenty-nine species, including *Porphyromonas gingivalis*, increased as the severity of disease increased, whereas 6 species, including *Rothia denticola*, decreased. The relative abundances of *P. gingivalis*, *Tannerella forsythia*, *Filifactor alocis*, and *Prevotella intermedia* determined by qPCR were significantly different among the groups. The three bacterial species *P. gingivalis*, *T. forsythia*, and *F. alocis* were positively correlated with the sum of the full-mouth probing depth and were moderately accurate at distinguishing the severity of periodontal disease. In conclusion, the salivary microbiota showed gradual compositional changes according to the severity of periodontitis, and the levels of *P. gingivalis*, *T. forsythia*, and *F. alocis* in mouth rinse saliva had the ability to distinguish the severity of periodontal disease.

**IMPORTANCE** Periodontal disease is one of the most widespread medical conditions and the leading cause of tooth loss, imposing high economic costs and an increasing burden worldwide as life expectancy increases. Changes in the subgingival bacterial community during the progression of periodontal disease can affect the entire oral ecosystem, and bacteria in saliva can reflect the degree of bacterial imbalance in the oral cavity. This study explored whether the specific bacterial species in saliva can distinguish the severity of periodontal disease by analyzing the salivary microbiota and suggested *P. gingivalis*, *T. forsythia*, and *F. alocis* as biomarkers for distinguishing the severity of periodontal disease in saliva.

**KEYWORDS** periodontal disease, saliva, microbiota, *Porphyromonas gingivalis*, *Tannerella forsythia*, *Filifactor alocis*

Address correspondence to Jae-Suk Jung, jsjung84@ajou.ac.kr.

The authors declare a conflict of interest. All authors are co-applicants on patent application 10-2021-0138897 entitled "Composition for diagnosing periodontal disease using microbiome in saliva and use thereof".

Periodontal disease is characterized by the inflammatory destruction of the periodontium in response to a multispecies bacterial community in the subgingival region. Gingivitis is an early inflammatory condition limited to the gingiva; however, if it is left untreated, it results in the progressive destruction of the periodontal supporting tissues, establishing periodontitis (1). This disease is one of the most widespread medical conditions and the leading cause of

tooth loss, imposing high economic costs (2) and an increasing burden worldwide as life expectancy increases. Moreover, scientists have increasingly recognized the importance of periodontal disease, which has been verified to be associated with various systemic diseases such as diabetes (3), rheumatoid arthritis (4), coronary heart disease (5), and Alzheimer's disease (6).

Dysbiotic changes in the microbiome have been verified in subgingival pockets during the progression of periodontal disease. The homeostatic balance between the symbiotic microbiota and the host immune response is maintained under healthy conditions, whereas the dysbiosis of the oral microbiota causes the inflammatory destruction of periodontal tissues (7). In the progression of periodontal disease, the host inflammatory reaction plays a pivotal role in the dysbiosis of the subgingival microbiota, and reciprocally, reinforced interactions between dysbiosis and inflammation drive chronic periodontitis (8). The entirety of the symbiotic and dysbiotic subgingival microbiota, rather than a single bacterium, affects the progression of periodontitis (9). Therefore, an understanding of the shift to a dysbiotic microbiota can provide useful information on the pathogenesis and diagnosis of the disease in addition to finding biomarkers indicative of the severity of periodontal disease and the progress of treatment.

Changes in the subgingival bacterial community can affect the entire oral ecosystem, and bacteria in saliva can reflect the degree of bacterial imbalance in the oral cavity because saliva contains microorganisms shed from various oral niches, including subgingival plaque. Studies have found a correlation between subgingival and salivary levels of specific bacteria present in periodontal tissues in patients with periodontitis (10–12). Those studies indicated that specific bacterial species in saliva could be used as a diagnostic tool for gauging the severity of periodontal disease. The use of saliva as a diagnostic tool for oral or systemic diseases has attracted attention because the collection of saliva is simple, rapid, and non-invasive (13, 14). In particular, saliva is useful as a near-patient tool for point-of-care diagnosis (15).

The objective of this study was to identify specific bacterial species to distinguish the severity of periodontal disease by analyzing the salivary microbiota using 16S rRNA gene sequencing and quantitative real-time PCR (qPCR). This approach can additionally reveal taxonomic compositional changes ranging from homeostatic balance under healthy conditions to dysbiosis indicating periodontal disease.

## RESULTS

**Sample groups and clinical responses to treatment.** Eight periodontally healthy subjects (H), 16 patients with gingivitis (G), 19 patients with moderate periodontitis (MP), and 29 patients with severe periodontitis (SP) were enrolled in this study. There were no significant differences in the mean ages ($P = 0.057$), numbers of teeth ($P = 0.07$), or numbers of teeth experiencing dental caries ($P = 0.177$), whereas the gender distribution ($P = 0.001$) and smoking status ($P = 0.001$) showed significant differences among the four groups. Two and eleven light smokers were included in the MP and SP groups, respectively. The clinical parameters of the four groups are detailed in Table 1. There were statistically significant differences in the sum of the plaque index (PI), the sum of the probing depth (PD), the mean PD, the sum of the modified sulcus bleeding index (mSBI), the sum of the gingival index (GI), the mean GI, the mean clinical attachment level (CAL), and mean bleeding on probing (BOP) (percent) of the full mouth among the four groups (Table 1).

**Taxon diversity of the four groups.** From a total of 72 bacterial communities, the average number of reads used for data analysis was 55,192 (5,314 to 102,131 per sample), with an average length of 421 bp and an average number of species per sample of 264. When alpha diversity metrics were applied using the number of identified species, the Chao1 index, the Shannon index, and phylogenetic diversity, the diversity tended to increase as the severity of periodontal disease increased (Fig. 1). The number of identified species in the H group was the lowest and significantly increased as the severity of periodontal disease increased, showing the highest number in the SP group. The results were similar using the Chao1 estimator and phylogenetic diversity, but there were no

**TABLE 1** Clinical characteristics of the entire mouth of grouped subjects[a]

| Characteristic | Value for group | | | | P value by a Mann-Whitney U test | P value by simple linear regression analysis |
| --- | --- | --- | --- | --- | --- | --- |
| | Healthy (n = 8) | Gingivitis (n = 16) | Moderate periodontitis (n = 19) | Severe periodontitis (n = 29) | | |
| General characteristics | | | | | | |
| No. of male/no. of female subjects | 2/6 | 0/16 | 9/10 | 17/12 | 0.001 | |
| Mean age (yrs) ± SD | 50.63 ± 16.43 | 43.44 ± 12.53 | 42.26 ± 12.34 | 50.76 ± 10.51 | 0.057 | |
| No. of nonsmokers/no. of current smokers | 8/0 | 16/0 | 17/2 | 18/11 | 0.001 | |
| | | | | | | |
| Oral characteristics | | | | | | |
| Mean sum of PI ± SD | 15.13 ± 11.90 | 34.94 ± 23.52 | 61.11 ± 23.85 | 61.86 ± 23.77 | <0.001 | <0.001 |
| Mean sum of PD ± SD | 315.50 ± 28.76 | 363.94 ± 38.14 | 430.68 ± 40.19 | 531.97 ± 95.58 | <0.001 | <0.001 |
| Mean PD ± SD | 1.92 ± 0.14 | 2.29 ± 0.25 | 2.63 ± 0.20 | 3.39 ± 0.63 | <0.001 | <0.001 |
| Mean sum of mSBI ± SD | 15.50 ± 0.53 | 106.75 ± 56.25 | 147.32 ± 49.48 | 178.10 ± 66.82 | <0.001 | <0.001 |
| Mean sum of GI ± SD | 79.63 ± 4.03 | 96.50 ± 13.80 | 109.26 ± 10.77 | 112.48 ± 16.86 | <0.001 | <0.001 |
| Mean GI ± SD | 1.46 ± 0.08 | 1.82 ± 0.25 | 2.00 ± 0.18 | 2.14 ± 0.27 | <0.001 | <0.001 |
| Mean CAL ± SD | 2.23 ± 0.26 | 2.37 ± 0.23 | 2.76 ± 0.18 | 3.98 ± 0.85 | <0.001 | <0.001 |
| Mean BOP (%) ± SD | 9.38 ± 0.32 | 51.57 ± 19.88 | 64.62 ± 11.88 | 71.77 ± 16.42 | <0.001 | <0.001 |
| Mean no. of teeth ± SD | 27.38 ± 0.74 | 26.56 ± 1.86 | 27.32 ± 1.29 | 26.28 ± 2.05 | 0.070 | |
| Mean no. of teeth experiencing dental caries ± SD | 3.38 ± 0.96 | 7.00 ± 1.11 | 5.16 ± 0.90 | 4.66 ± 1.10 | 0.177 | |

[a]Values are presented as the means ± standard deviations. PI, plaque index; PD, probing depth; mSBI, modified sulcus bleeding index; GI, gingival index; CAL, clinical attachment level; BOP, bleeding on probing. A P value of <0.05 was considered to indicate statistical significance.

significant differences between the H and G groups. The Shannon index showed a significant difference between the H and MP groups.

**Differences in taxon distributions among the four groups.** To investigate the differences in the composition of the salivary microbiota among the groups, the relative abundances of the taxa were compared at the phylum, genus, and species levels (Fig. 2). The distribution patterns of the top 6 phyla, except those with an abundance of <1%, in each group are shown in Fig. 2a. In all groups, *Firmicutes* and *Proteobacteria* were the highest-abundance phyla in the salivary microbiome, accounting for approximately 55 to 70% of the total bacteria; however, these were not significantly different among groups. *Bacteroidetes*, *Fusobacteria*, and *Spirochaetes* were significantly more abundant as the severity of periodontal disease increased, while *Actinobacteria* were significantly more abundant in the healthy

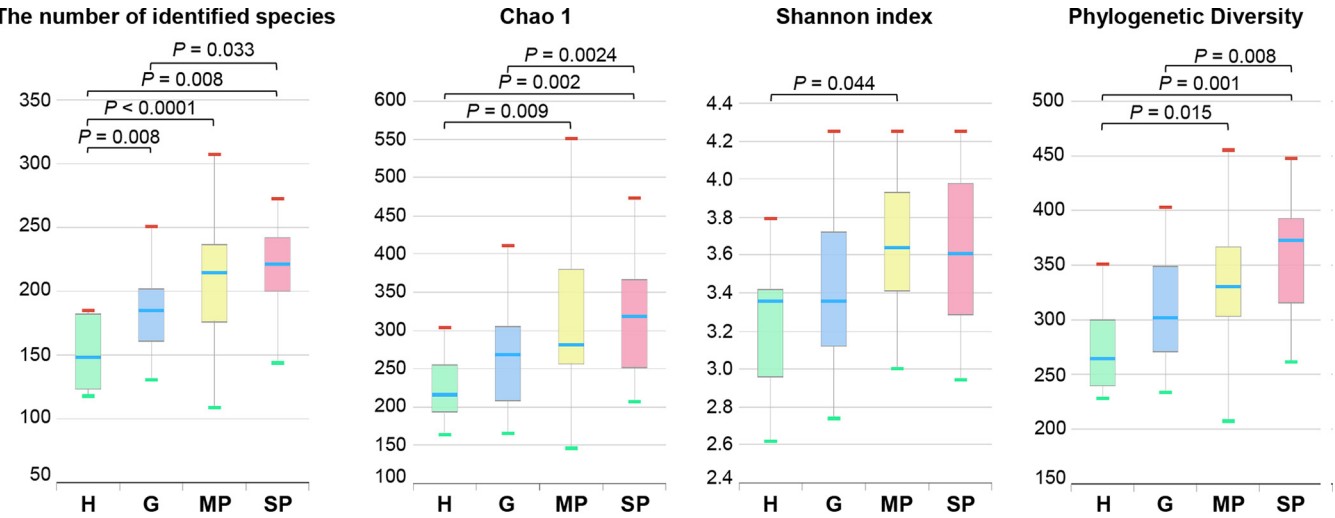

**FIG 1** Comparison of the numbers of identified species, Chao1 indices, Shannon indices, and phylogenetic diversities. Saliva samples were collected from 8 periodontally healthy subjects (H group), 16 patients with gingivitis, 19 patients with moderate periodontitis (MP group), and 29 patients with severe periodontitis (SP group). Each value is presented as a box plot. The top, middle, and bottom lines of the boxes represent the 25th, 50th (median), and 75th percentiles, respectively. The significance of each difference among the four groups was evaluated using the Wilcoxon rank sum test, and a P value of <0.05 was considered to indicate statistical significance. Only P values of <0.05 are indicated.

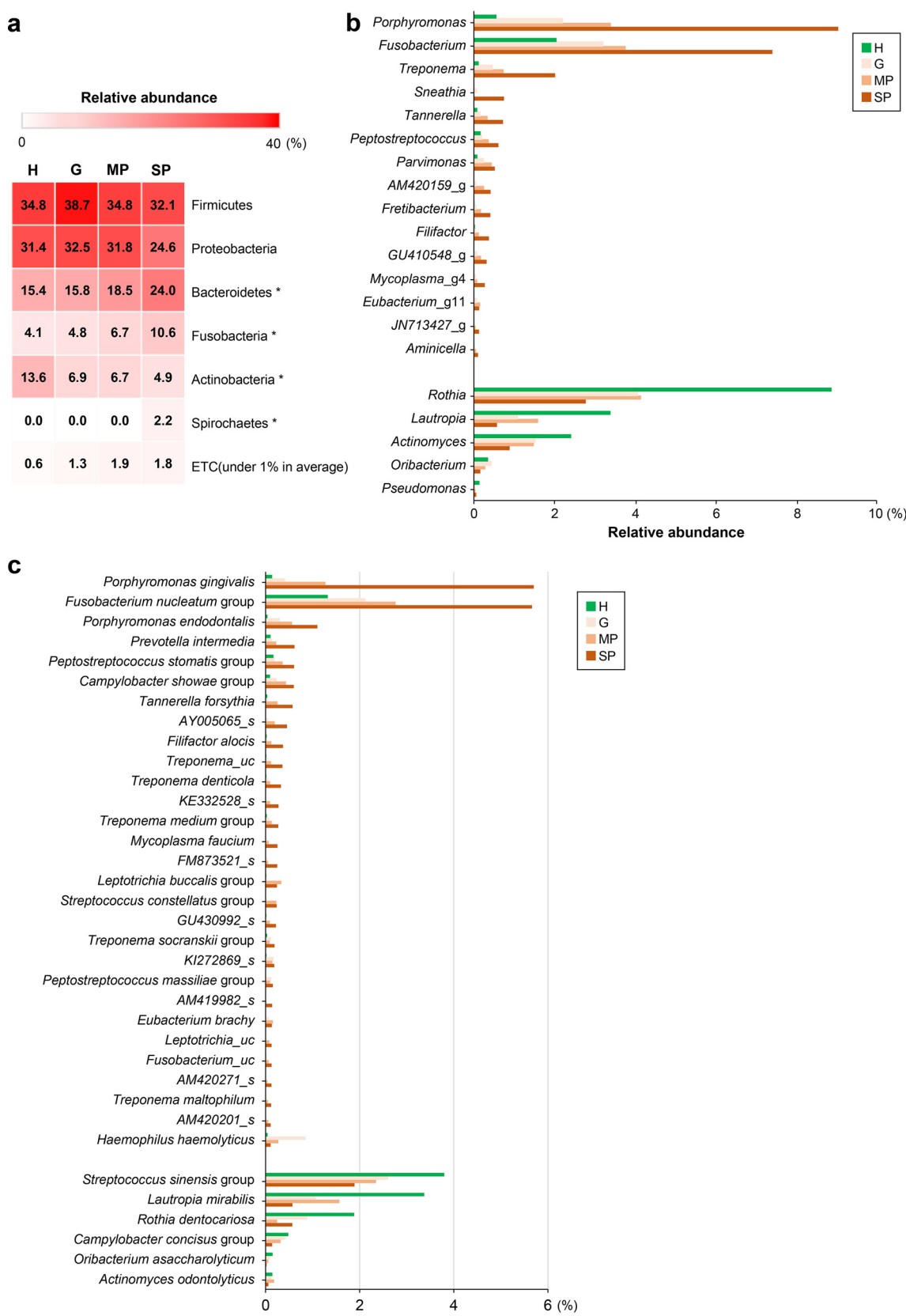

**FIG 2** Differences in taxon distributions among the four groups at the phylum, genus, and species levels. (a) Distribution patterns of the top 6 phyla, except for phyla with an abundance of <1% on average, in each sample group. *, $P < 0.05$ (by a Kruskal-Wallis H test

group. Among the genera and species with a >0.1% relative abundance in saliva, 15 and 29 had significantly increased abundances as the severity of periodontal disease increased, respectively. In contrast, 5 genera and 6 species were more abundant in the H group. Among the taxa more abundant in periodontal disease, the top 7 genera were *Porphyromonas*, *Fusobacterium*, *Treponema*, *Sheathia*, *Tannerella*, *Peptostreptococcus*, and *Parvimonas*, and the top 7 species were *Porphyromonas gingivalis*, the *Fusobacterium nucleatum* group, *Porphyromonas endodontalis*, *Prevotella intermedia*, the *Peptostreptococcus stomatis* group, the *Campylobacter showae* group, and *Tannerella forsythia*. The abundances of *Treponema denticola*, one of the red complex bacteria (16), and *Filifactor alocis*, a relatively newly identified periodontopathogen (17), were also significantly increased as the severity of periodontal disease increased. The top 3 genera and species in the healthy groups were the genera *Rothia*, *Lautropia*, and *Actinomyces* and the species *Streptococcus sinensis* group, *Lautropia mirabilis*, and *Rothia dentocariosa*.

**qPCR of specific bacterial species.** qPCR was performed for the periodontitis-associated species *P. gingivalis*, *T. forsythia*, *T. denticola*, *P. intermedia*, *P. endodontalis*, *F. alocis*, and *F. nucleatum* and the health-associated species *R. dentocariosa*, which showed significant differences by 16S rRNA gene sequencing. Among the top 11 periodontitis-related species, the *Peptostreptococcus stomatis* group (https://www.ezbiocloud.net/mtp/taxonomy?db=PKSSU4.0&tn=Peptostreptococcus%20stomatis%20group&depth=2&rg=undefined) and the *Campylobacter showae* group (https://www.ezbiocloud.net/mtp/taxonomy?db=PKSSU4.0&tn=Campylobacter%20showae%20group&depth=2&rg=undefined) were excluded because they are not a single species. Although *F. nucleatum* is classified as the *F. nucleatum* group by 16S rRNA gene sequencing, qPCR was performed because of its importance for biofilm formation and microbial structures. *Parvimonas micra* was additionally analyzed as a periodontitis-associated bacterial species because its distribution in periodontitis was decreased following nonsurgical periodontal treatment in our previous study (18). Among the top 3 health-related species, only *R. dentocariosa* was analyzed by qRT-PCR because the *S. sinensis* group is not a single species (https://www.ezbiocloud.net/mtp/taxonomy?db=PKSSU4.0&tn=Streptococcus%20sinensis%20group&depth=2&rg=v3v4), and *L. mirabilis* was detected in only a few samples when analyzed by using a newly designed primer (data not shown).

The bacterial count was calculated as the number of bacteria in 11.43 $\mu$L of a mouth rinse saliva sample, considering that 2 $\mu$L of the DNA template was used for qPCR, and the relative abundance of bacterial species was calculated as the ratio of the specific bacterial count to the total bacterial count. The correlation between the percent relative abundance determined by 16S rRNA gene sequencing and the percent relative abundance determined by qPCR [qPCR(%)] was analyzed for each bacterial species. A significant positive correlation was found for all bacterial species (Fig. 3). However, there were some differences in the correlation coefficient ($R^2$) values. The $R^2$ values of *P. gingivalis*, *T. forsythia*, *P. intermedia*, and *F. alocis* were >0.7, showing relatively strong correlations. *F. nucleatum* showed the lowest $R^2$ value, probably because the group was analyzed using 16S rRNA gene sequencing that included several species of *F. nucleatum*, *F. polymorphum*, *F. vincentii*, *F. animalis*, *F. simiae*, *F. canifelinum*, and *F. hwasookii* (https://www.ezbiocloud.net/mtp/taxonomy?db=PKSSU4.0&tn=Fusobacterium%20nucleatum%20group&rg=V3V4). Based on the genome-based approach, *F. nucleatum* subsp. *nucleatum*, *F. nucleatum* subsp. *polymorphum*, *F. nucleatum* subsp. *vincentii*, and *F. nucleatum* subsp. *animalis* were classified as *F. nucleatum*, *F. polymorphum*, *F. vincentii*, and *F. animalis*, respectively (19). The qPCR(%) values for *P. gingivalis*,

**FIG 2** Legend (Continued)

among the four groups). (b) Genera that were significantly different among the four groups among those with a >0.1% relative abundance in saliva. As the severity of periodontal disease increased, genera that were more abundant are in the top 15 of the graph, whereas genera that were more abundant in the healthy controls are in the bottom 5. (c) Species significantly different among the four groups among genera with a >0.1% relative abundance in saliva. As the severity of periodontal disease increased, species that were more abundant are in the top 29 of the graph, whereas species that were more abundant in the healthy controls are in the bottom 6. A P value of <0.05 was considered statistically significant by the Kruskal-Wallis H test. H, healthy group; G, gingivitis group; MP, moderate periodontitis group; SP, severe periodontitis group; _g: the genus name was unknown; _s, the species name was unknown.

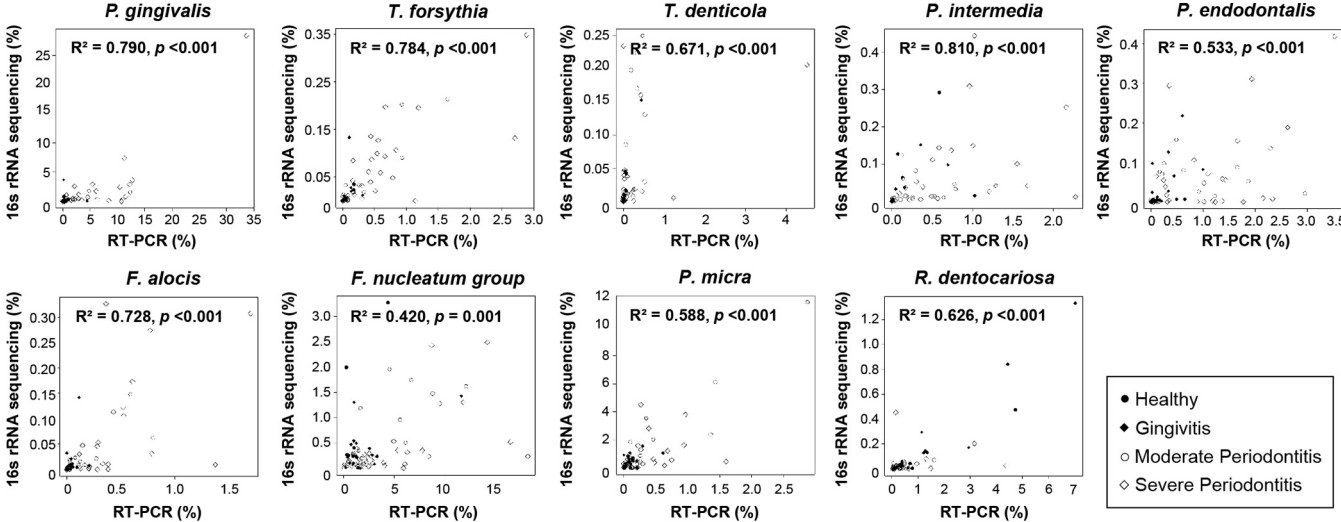

**FIG 3** Correlation between the relative abundances of 9 bacterial species determined by 16S rRNA gene sequencing and those determined by qPCR. The correlation analysis was established by comparing the relative abundances of the 9 species established using 16S rRNA gene sequencing to the relative abundance obtained using qPCR from a total of 72 saliva samples from the four groups. The scatterplots were generated using the Spearman method.

*T. forsythia*, *P. intermedia*, and *F. alocis* showed significant differences among the groups ($P < 0.01$). Simple linear regression analysis showed that the qPCR(%) of *R. dentocariosa* decreased as the severity of disease increased ($P < 0.01$) (Fig. 4a). For bacterial counts determined by qPCR [qPCR(count)], all bacterial species except *R. dentocariosa* were significantly different among the groups and increased as the severity of disease increased (Fig. 4b).

Next, we analyzed whether the qPCR(%) or qPCR(count) of bacterial species was correlated with the sum of PD. The sum of PD significantly increased with the increasing severity of disease (Table 1). The qPCR(%) or qPCR(count) of all bacterial species except *R. dentocariosa* showed a tendency toward a positive correlation with the sum of PD. In common, *P. gingivalis*, *T. forsythia*, and *F. alocis* showed relatively high $R^2$ values of >0.3 (Fig. 5).

The prediction performance of nine bacterial species in distinguishing the severity of periodontal disease was evaluated using a receiver operating characteristic (ROC) curve. *P. gingivalis* showed an area under the ROC curve (AUC) of 0.73 to 0.82 and a sensitivity and specificity of >72% for all divisions (Table 2). In division 1 (D1), the qPCR(%) and qPCR (count) values of *F. alocis* and *P. endodontalis* showed AUC values of >0.8. Especially, qPCR (%) of 0.004 for *F. alocis* distinguished the healthy group with a sensitivity of 0.84 and a specificity of 0.88, and five of the qPCR(count) values for *F. alocis* distinguished the healthy group with a sensitivity of 0.89 and a specificity of 0.88. In D3, the qPCR(count) values for *T. forsythia* and *P. intermedia* showed relatively high AUC values of >0.8. A qPCR(count) of 213 for *T. forsythia* distinguished the SP group with a sensitivity of 0.62 and a specificity of 0.98 (Table 2).

## DISCUSSION

This study explored whether the specific bacterial species in saliva can distinguish the severity of periodontal disease by analyzing the salivary microbiota using 16S rRNA gene sequencing and qPCR. As a result, *P. gingivalis*, *T. forsythia*, and *F. alocis* were especially superior to other bacterial species for the diagnosis of disease based on the differences among the four groups, the correlation with the sum of PD, and the prediction performance of each bacterial species. The strong association of salivary *P. gingivalis* with periodontal disease and its excellent ability to distinguish the severity of periodontal disease have been consistently found in other studies (20, 21). A large-scale study comprising 977 Japanese individuals demonstrated that the salivary level of *P. gingivalis* correlates with the percentage of sites with a probing pocket depth of ≥4 mm (20). Moreover, a sequencing-based study showed that the relative abundance of *P. gingivalis* could discriminate patients with periodontitis from orally healthy controls with an AUC of 0.80 (21). The

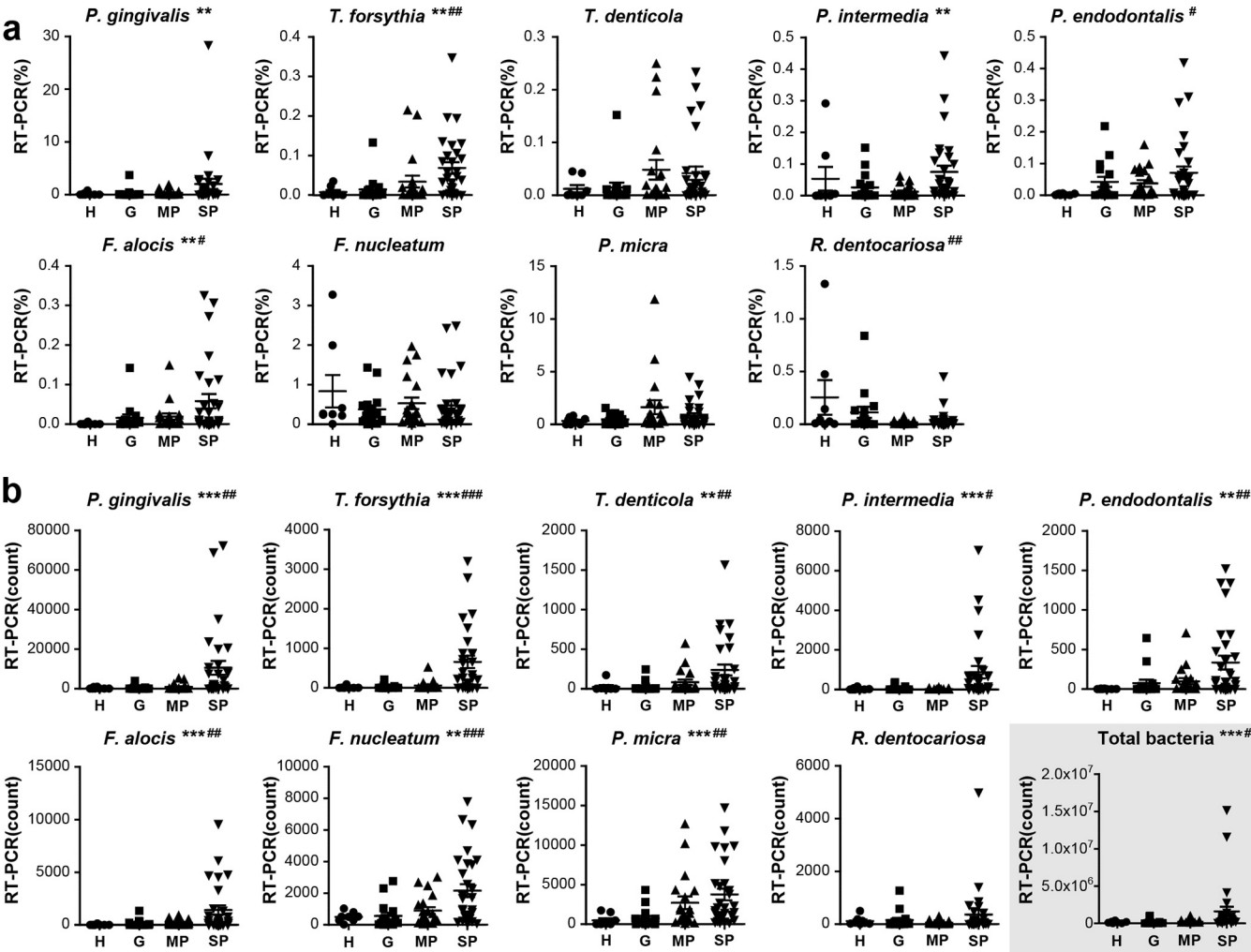

**FIG 4** Comparison of the relative abundances and bacterial counts of 9 bacterial species determined by qPCR. Saliva samples were collected from 8 periodontally healthy subjects (H group), 16 patients with gingivitis, 19 patients with moderate periodontitis (MP group), and 29 patients with severe periodontitis (SP group). (a) Comparison of the relative abundances of 9 bacterial species determined by qPCR among the four groups. (b) Comparison of the bacterial counts of 9 bacterial species determined by qPCR among the four groups. *, $P < 0.05$; **, $P < 0.01$; ***, $P < 0.001$ (by a Kruskal-Wallis H test among the four groups). #, $P < 0.05$; ##, $P < 0.01$; ###, $P < 0.001$ (by simple linear regression analysis among the four groups).

excellence of *F. alocis* and *T. forsythia* in distinguishing the severity of periodontal disease was supported by a recent study of a Swedish cohort showing a significant periodontitis-associated microbiota with increased levels of *T. forsythia*, *F. alocis*, and *P. micra* (22). Across all studies, however, the ability of the nine species of interest in this study to distinguish between health and disease is not consistently superior to those of other bacterial species. A recent study using qPCR results for 11 bacterial species, including 6 species analyzed in this study, resulted in the determination that the combination including *P. micra* demonstrated significantly higher AUC values for relatively mild criteria of disease, while combinations including *Streptococcus constellatus*, *P. gingivalis*, and *F. nucleatum* subsp. *vincentii* demonstrated significantly higher AUC values for detection (23). The results regarding *P. gingivalis* abundance are consistent with our results; however, *T. forsythia* and *F. alocis* did not reflect the severity of disease compared to the four above-mentioned bacteria (23). Another recent study analyzing the salivary microbiota of a total of 45 subjects using full-length bacterial 16S rRNA gene sequencing suggested that higher abundances of *P. intermedia* and *Catonella morbi* and lower abundances of *Porphyromonas pasteri*, *Prevotella nanceiensis*, and *Haemophilus parainfluenzae* might be biomarkers of periodontitis, with an AUC reaching 0.9733 (24). With the exception of *P. intermedia*, no bacterial species showed a significant difference in this study. These differences among studies may be caused by ethnic or geographic differences in

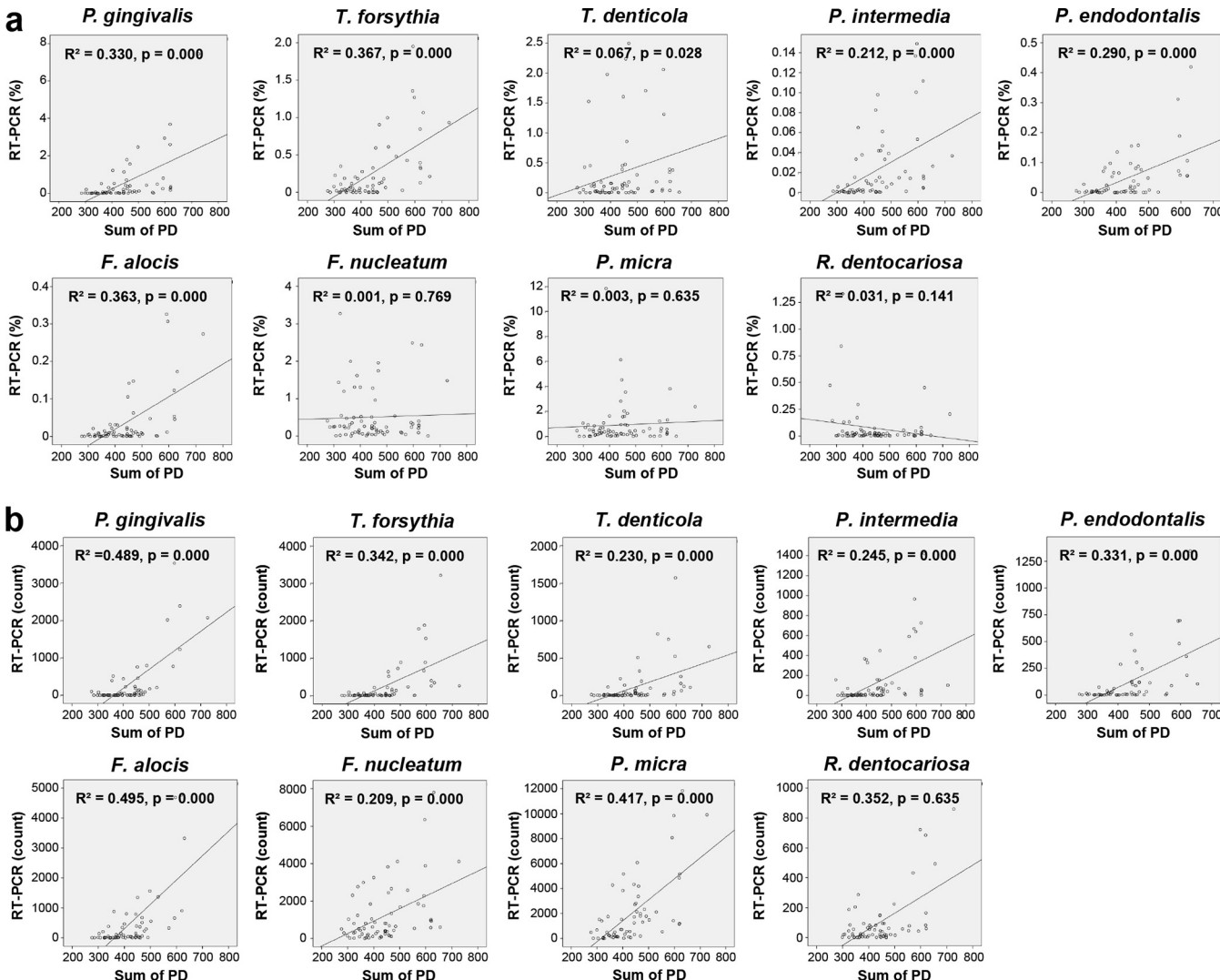

**FIG 5** Correlations between the levels of the 9 bacterial species and the sum of the full-mouth probing depth. The scatterplots were generated using simple linear regression analysis. The correlation analysis was established by comparing the levels of 9 bacterial species determined by qPCR in saliva samples to the sum of the full-mouth probing depth from 72 subjects of the four groups. (a) Correlation between the relative abundance (percent) of 9 bacterial species established using qPCR and the sum of the full-mouth PD. (b) The correlation between the bacterial counts of 9 bacterial species was established using qPCR and the sum of the full-mouth PD.

subjects, saliva sampling methods, nucleic acid extraction methods, bias of sequencing methods, diet, bacterial detection methods, local genetic differences in bacteria, or the pathogenicity of bacterial strains. These results suggest that *P. gingivalis*, *T. forsythia*, and *F. alocis* might be biomarkers for the severity of periodontal disease in Koreans. However, the prediction performance of the nine bacterial species in distinguishing the severity of periodontal disease should be investigated in a distinct large population to confirm external validity.

In the present study, *T. denticola*, one of the red complex bacterial species, showed a significant difference among the groups by 16S rRNA gene sequencing; however, the relative abundance by qPCR was not significantly different among groups (Fig. 4). Moreover, the $R^2$ value in correlation with the sum of PD and the AUC level for prediction performance were relatively lower than those of the other red complex bacteria (Fig. 5 and Table 2). Therefore, the association of *T. denticola* with periodontal disease was suggested to be relatively weaker than those for the other red complex bacteria. This was an exceptional finding because the association of red complex bacteria in subgingival plaque with periodontitis has been well identified (16, 25). However, this finding has something in common with a recent study of Japanese people examining the association of salivary bacterial species

**TABLE 2** Cutoff value, AUC, sensitivity, and specificity of each bacterial species established by qPCR to distinguish the severity of periodontal disease

| Species | qPCR(%) | | | | qPCR(count) | | | |
|---|---|---|---|---|---|---|---|---|
| | Cutoff | AUC | Sensitivity | Specificity | Cutoff | AUC | Sensitivity | Specificity |
| **D1** | | | | | | | | |
| P. gingivalis | 0.0034 | 0.79 | 0.83 | 0.75 | 5 | 0.82 | 0.89 | 0.75 |
| T. forsythia | 0.0021 | 0.77 | 0.78 | 0.75 | 10 | 0.75 | 0.63 | 0.88 |
| T. denticola | 0.0000 | 0.69 | 0.88 | 0.50 | 11 | 0.73 | 0.58 | 0.88 |
| P. intermedia | 0.0013 | 0.71 | 0.80 | 0.63 | 1 | 0.71 | 0.80 | 0.63 |
| P. endodontalis | 0.0062 | 0.80 | 0.59 | 1.00 | 9 | 0.82 | 0.64 | 1.00 |
| F. alocis | 0.0004 | 0.86 | 0.84 | 0.88 | 5 | 0.88 | 0.89 | 0.88 |
| F. nucleatum | 0.2401 | 0.63 | 0.48 | 0.25 | 1036 | 0.68 | 0.36 | 1.00 |
| P. micra | 0.0077 | 0.67 | 0.97 | 0.38 | 94 | 0.78 | 0.94 | 0.63 |
| R. dentocariosa | 0.0269 | 0.66 | 0.31 | 0.38 | 44 | 0.56 | 0.50 | 0.38 |
| | | | | | | | | |
| **D2** | | | | | | | | |
| P. gingivalis | 0.0586 | 0.73 | 0.71 | 0.75 | 33 | 0.78 | 0.81 | 0.75 |
| T. forsythia | 0.0048 | 0.71 | 0.75 | 0.67 | 7 | 0.77 | 0.79 | 0.75 |
| T. denticola | 0.0113 | 0.66 | 0.52 | 0.79 | 11 | 0.77 | 0.71 | 0.83 |
| P. intermedia | 0.0037 | 0.68 | 0.73 | 0.63 | 3 | 0.71 | 0.79 | 0.63 |
| P. endodontalis | 0.0074 | 0.63 | 0.58 | 0.67 | 29 | 0.71 | 0.58 | 0.83 |
| F. alocis | 0.0062 | 0.68 | 0.60 | 0.75 | 16 | 0.75 | 0.83 | 0.67 |
| F. nucleatum | 0.2370 | 0.56 | 0.50 | 0.38 | 529 | 0.71 | 0.67 | 0.75 |
| P. micra | 1.1486 | 0.61 | 0.27 | 0.96 | 233 | 0.74 | 0.90 | 0.58 |
| R. dentocariosa | 0.0269 | 0.68 | 0.23 | 0.42 | 11 | 0.57 | 0.81 | 0.33 |
| | | | | | | | | |
| **D3** | | | | | | | | |
| P. gingivalis | 0.1122 | 0.74 | 0.76 | 0.72 | 838 | 0.78 | 0.72 | 0.84 |
| T. forsythia | 0.0297 | 0.74 | 0.62 | 0.86 | 213 | 0.80 | 0.62 | 0.98 |
| T. denticola | 0.0125 | 0.61 | 0.52 | 0.70 | 25 | 0.71 | 0.66 | 0.77 |
| P. intermedia | 0.0037 | 0.74 | 0.90 | 0.58 | 18 | 0.80 | 0.86 | 0.74 |
| P. endodontalis | 0.0329 | 0.60 | 0.48 | 0.72 | 79 | 0.69 | 0.59 | 0.79 |
| F. alocis | 0.0267 | 0.66 | 0.41 | 0.91 | 159 | 0.73 | 0.69 | 0.77 |
| F. nucleatum | 0.2401 | 0.55 | 0.45 | 0.44 | 529 | 0.72 | 0.79 | 0.65 |
| P. micra | 1.1486 | 0.57 | 0.28 | 0.86 | 891 | 0.69 | 0.76 | 0.63 |
| R. dentocariosa | 0.0167 | 0.66 | 0.28 | 0.40 | 215 | 0.59 | 0.28 | 0.91 |

counts and the severity of periodontal disease. As the *T. denticola* count increased, the rate of subjects with severe PD or bone loss also increased; however, this tendency was not as strong as that for *P. gingivalis* (20). It is not clear whether the relatively weak association of *T. denticola* with disease is because subgingival *T. denticola* was not well reflected in saliva, unlike other red complex bacterial species, or whether it is a characteristic specific to Koreans or Asians. Further studies are needed to simultaneously analyze the subgingival and salivary microbiota in a large population, perhaps using more genetic markers. These findings indicate that major periodontal pathogens could vary by ethnicity or geography.

16S rRNA gene sequencing revealed that the distinct microbiota of saliva tends to gradually diversify with the transition from health to severe periodontitis. Analysis of the relative abundances at the phylum level showed that *Bacteroidetes*, *Fusobacteria*, and *Spirochaetes* were significantly more abundant as the severity of periodontal disease increased, while *Actinobacteria* were significantly more abundant in the healthy group. These results clearly indicate the gradual change of the salivary microbiota to dysbiosis associated with disease progression. The increase in diversity with the increasing severity of disease was consistent with previous findings reported for plaque samples from patients with periodontitis and a healthy control group (26, 27). The increased abundances of *Bacteroidetes*, *Fusobacteria*, and *Spirochaetes* and the decreased abundance of *Actinobacteria* with the increasing severity of disease are consistent with previous findings in plaque samples (17, 25, 28). Regarding the phylum distributions in the salivary microbiota, a recent report showed that *Actinobacteria*, *Spirochaetes*, *Synergistetes*, and *Tenericutes* were more abundant in samples from patients with periodontitis (22). The results of that study regarding *Actinobacteria* were in contrast to our findings as well as previous findings obtained in

plaque samples (25, 28). In this study, the proportion of *Actinobacteria* in the H group (13.6%) was almost double those in the G, MP, and SP groups (4.9 to 6.9%). The difference between the H and G groups is based on the difference in full-mouth BOP percentages and not bone destruction. Our finding for the phylum *Actinobacteria*, which reflects the distribution of subgingival plaque, might be because the H group was strictly divided as subjects with a total oral BOP of <10% of the full-mouth BOP according to the newly classified periodontal disease classification system (29). The two phyla *Synergistetes* and *Tenericutes* had abundances of <1% in our findings, so they are not presented in Fig. 2. However, these two phyla were significantly different among the groups, showing a higher percentage as the severity of periodontal disease increased (see Fig. S2 in the supplemental material). These results reveal that the compositional diversity and distribution of phyla shown in subgingival plaque are reflected in the salivary microbiota accompanying gradual changes as the severity of periodontal disease increases.

In the distribution of subjects, gender and smoking status showed significant differences among the four groups. To limit the effect of the imbalanced distribution, 16S rRNA gene sequencing was performed for a total of 43 nonsmoking females (Fig. S3). The differences found at the phylum, genus, and species levels were almost identical to the results for the 72 total participants (Fig. S3). When analyzing only nonsmokers, the relative abundance of 9 bacterial species analyzed by qPCR was almost identical to that for the 72 total subjects (Table S4). The relative abundances of 9 bacterial species were not significantly different between nonsmokers and smokers in the SP group or between males and females in the MP and SP groups (Tables S5 to S7). These results are in line with the conclusion that smoking and gender have a limited association with bacterial activity compared to oral health status (30). However, since smoking has been considered one of the factors that affect the oral microbiota (31, 32), additional studies are needed.

In conclusion, this study suggests *P. gingivalis*, *T. forsythia*, and *F. alocis* as biomarkers for distinguishing the severity of periodontal disease in saliva. As the severity of periodontal disease increased, the phyla *Bacteroidetes*, *Fusobacteria*, and *Spirochaetes* were significantly more abundant, while the phylum *Actinobacteria* was significantly less abundant. This suggestion is based on the differences in distributions among the four groups, the correlation with the sum of PD, and the prediction performance of each bacterial species. However, large-scale studies are needed to determine whether these three bacterial species can distinguish all stages of the disease and to determine the best optimal values for differentiating the severity of disease.

## MATERIALS AND METHODS

**Subjects.** All participants were recruited from Ajou University Dental Hospital from May 2018 to March 2020. This study was approved by the Institutional Review Board for Human Subjects of Ajou University Dental Hospital (approval number AJOUIRB-SMP-2018-062). Informed consent was obtained from all subjects. The participants had no history of systemic disease that could influence the prognosis of periodontitis, untreated caries, or orthodontic appliances. None of the subjects were pregnant/breast-feeding, heavy smokers smoking >10 cigarettes per day, or treated with antibiotics, antimicrobials, and/or anti-inflammatory drugs during the 3 months prior to examination and sampling. All participants received a full-mouth dental examination after the sampling of saliva. One specialized periodontist performed the periodontal examinations of all participants using manual periodontal probes (Pcpunc 15; HuFriedy Manufacturing Co., Inc., Chicago, IL, USA). The clinical periodontal indices plaque index (PI), gingival index (GI), probing depth (PD), clinical attachment level (CAL), bleeding on probing (BOP), and modified sulcus bleeding index (mSBI) were recorded. Participants were classified into four groups, periodontally healthy controls (H group), gingivitis (G group), moderate periodontitis (MP group), and severe periodontitis (SP group), based on their periodontal status according to the clinical criteria stated in the consensus report of the World Workshop on Periodontics (29). The subjects in the H group exhibited no sites with attachment loss, no sites with a PD of >3 mm and a BOP value of <10%, and no radiographic alveolar bone loss. The patients in the G group exhibited no clinical attachment loss, no sites with a PD of >3 mm and a BOP value of ≥10%, and no radiographic alveolar bone loss. The MP group included individuals with a PD of ≤5 mm, almost horizontal bone loss, no experience of tooth extraction due to periodontitis, and alveolar bone resorption limited to the coronal third (~15% to 33%). The SP group included individuals exhibiting a local PD of ≥6 mm, vertical bone loss of ≥3 mm, and molars characterizing furcation involvement (class II or III).

**Mouth rinse saliva sampling.** Saliva was collected by the mouth rinse method to obtain more bacteria by washing away ones attached to oral surfaces, including teeth, through gargling. Previous studies showed that this saliva collection method had a minimal effect on salivary microbial community profiles,

and mouth rinse samples performed similarly to unstimulated saliva samples for analysis of the oral microbiome (33–35). All participants were instructed to prohibit ingestion, rinsing, and oral hygiene measures for 1 h prior to sampling. The saliva samples were collected after gargling for 1 min with a 10-mL normal saline solution and then transferred to the laboratory on ice as soon as possible. From each sample, 8 mL and 200 $\mu$L were fractionated for 16S rRNA sequencing and qPCR, respectively. The samples were centrifuged at 15,928 $\times$ $g$ for 5 min, and the pellets were stored at $-80°C$ until DNA extraction.

**DNA extraction, amplification of the 16S rRNA gene, and Illumina sequencing.** Within a few days, the pellets were transferred to ChunLab, Inc. (https://www.cjbioscience.com/), for DNA extraction and sequencing analysis. DNA was extracted using a FastDNA spin kit for soil (MPBio), and PCR amplification was performed using primers targeting the V3 and V4 regions of the 16S rRNA gene. The PCR products were sequenced with an Illumina MiSeq sequencing system at ChunLab, Inc., using UCHIME (36). Data from public sources, including the Human Microbiome Project and the Sequence Read Archive (SRA), were processed in advance. Sequencing raw data (in FASTQ or FASTA format) were uploaded to EzBioCloud (https://www.ezbiocloud.net), and the nonchimeric 16S rRNA database from EzBioCloud was used to detect chimeric sequences in reads that contained a <97% best-hit similarity rate. We generated taxonomic profiles from the sequencing data and grouped and compared the profiles from different samples. The bacterial DNA for qPCR was sent back to the Ajou University laboratory in a storage container at 4°C.

**qPCR for oral bacterial species.** Sequences of >97% identity represent the same species in sequencing analysis, and partial variable regions of the 16S rRNA gene for sequencing make it impossible to obtain accurate information at the species level (37). Therefore, qPCR was performed on bacterial species showing statistically significant differences among the four groups by 16S rRNA gene sequencing analysis. To identify the bacterial counts and the relative abundances compared to the total bacteria in saliva samples, qPCR analysis was performed on the total bacteria and 9 bacterial species using a universal primer and species-specific primers, respectively (see Table S1 in the supplemental material). The analysis of total bacteria, *P. micra*, and *F. alocis* was performed using primers targeting the 16S rRNA gene, and the value obtained by qPCR was divided by the count of the copy number of the 16S rRNA gene in one bacterium (38, 39). The other 7 species were identified using primers designed based on the nucleotide sequence of the RNA polymerase $\beta$-subunit gene (*rpoB*), and quantitative analysis was performed based on the formula deduced from previous studies (40–42). qPCR was performed with AccuPower GreenStar qPCR PreMix using an Exicycler 96 real-time quantitative thermal block (Bioneer). Each PCR was performed with a total volume of 50 $\mu$L containing 3 $\mu$L each of the forward and reverse primers (with final concentrations of 500 nM each), 2 $\mu$L of genomic DNA, and the appropriate dose of sterilized DNase-RNase-free water in PreMix PCR tubes. The qPCR conditions were an initial denaturation step at 95°C for 10 min followed by 40 cycles of denaturation at 95°C for 10 s, primer annealing and extension at 72°C for 30 s, and a final cooling step at 25°C for 1 min. The reaction specificities were confirmed by melting-curve analysis with a progressive increase in the temperature from 65°C to 94°C at a 1°C/s transition rate and continuous fluorescence acquisition. For quantitative analysis of *F. alocis* and *P. micra*, absolute quantification of the DNA was performed using the standard curve method. First, *F. alocis* (KCOM 3031) and *P. micra* (KCOM 1533) were grown in brain heart infusion (BHI) broth (catalog number 237500; BD) supplemented with 0.5% yeast extract, 0.05% cysteine HCl–H$_2$O, 0.5 mg/mL hemin, and 2 $\mu$g/mL vitamin K$_1$ at 37°C in the anaerobic pouch (catalog number 260683; BD). Both bacterial species were used to generate the standard curves and as positive controls. Regression analysis was performed to obtain the equations used to interpolate the threshold cycle ($C_T$) values from samples and quantify the corresponding concentrations of genomic DNA of each bacterium in each sample. The $R^2$ value was used as a measure of the goodness of fit of the regression analysis. Based on the equation, the bacterial counts of *F. alocis* and *P. micra* were calculated. A standard curve was obtained using the average from three independent experiments.

**Statistical analysis.** The sums of the full-mouth PI, PD, mSBI, and GI; the means of the PD, GI, and clinical CAL; and the percentage of BOP were compared among the four groups using the Kruskal-Wallis test. Analyses of the microbiome taxonomic profiles and comparisons among the four groups were performed using BIOiPLUG (https://www.ezbiocloud.net), a Web-based life information analysis cloud platform provided by ChunLab, Inc. The analysis was based on the relative abundance of each taxonomic group, and the results were reconstructed for this study. Comparisons of the numbers of identified species, the Chao1 indices, the Shannon indices, and the phylogenetic diversities among the four groups were performed using the Wilcoxon rank sum test. The Kruskal-Wallis test was conducted to evaluate the differences in the dominant operational taxonomic units (OTUs) among the four groups. The correlations between the relative abundances of specific bacterial species determined by 16S rRNA gene sequencing and those determined by qPCR were analyzed using Spearman correlations. The correlations between the bacterial level describing the relative abundance determined by qPCR [qPCR(%)] and the bacterial count determined by qPCR [qPCR(count)] and the sum of full-mouth PD were analyzed using a simple linear regression model. The comparison of qPCR(%) and qPCR(count) values among the four groups was performed using the Kruskal-Wallis test. A simple linear regression model was used to analyze whether values tended to increase (or decrease) as the severity of periodontal disease increased. The prediction performance of 9 bacterial species in distinguishing the severity of periodontal disease was evaluated using a receiver operating characteristic (ROC) curve. The area under the ROC curve was calculated to compare the predictive abilities of the indices. The cutoff value for each bacterial species was calculated to distinguish the severity of periodontal disease with 3 divisions (D1 to D3). D1 was used to distinguish healthy conditions (H group) from periodontal disease (G, MP, and SP groups), D2 was used to distinguish the condition of no bone loss (H and G groups) from periodontitis (MP and SP groups), and D3 was used to distinguish severe periodontitis from the other conditions (H, G, and MP groups). The optimal cutoff values were determined to maximize the AUC value, and the sensitivity and specificity were then analyzed. Based on the AUC statistic, the diagnostic test can be either noninformative (AUC = 0.5), less accurate (0.5 < AUC $\leq$ 0.7), moderately accurate (0.7 < AUC $\leq$ 0.9), highly accurate (0.9 < AUC < 1), or perfect (AUC = 1) (43). All of the

statistical analyses were performed using SAS (version 9.4; SAS Institute, Inc., Cary, NC, USA) and the R package (version 4.1.2; R Foundation for Statistical Computing, Vienna, Austria). The results were considered statistically significant when the $P$ values were <0.05.

**Data availability.** The data for this study have been deposited in the European Nucleotide Archive (ENA) at EMBL-EBI under accession number PRJEB61123.

## SUPPLEMENTAL MATERIAL

Supplemental material is available online only.
**SUPPLEMENTAL FILE 1**, PDF file, 0.7 MB.

## ACKNOWLEDGMENTS

This work was supported by a National Research Foundation of Korea (NRF) grant funded by the South Korean government (MSIT) (number 2018R1C1B6009545).

All authors are coapplicants on patent application 10-2021-0138897 entitled "Composition for diagnosing periodontal disease using microbiome in saliva and use thereof."

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
