## [Reviewer comments · Microbiology Spectrum]

Microbiology Spectrum

The characteristics of salivary microbiota in periodontal diseases and the potential roles of individual bacterial species to predict the severity of periodontal disease

Suk Ji, Joong-Ki Kook, Jae-Suk Jung, Soon-Nang Park, Yun Kyong Lim, and Geum Hee Choi

Corresponding Author(s): Jae-Suk Jung, Ajou University Hospital

Review Timeline:

Submission Date:	October 28, 2022
Editorial Decision:	January 31, 2023
Revision Received:	April 21, 2023
Accepted:	April 27, 2023

Editor: Justin Kaspar

Reviewer(s): Disclosure of reviewer identity is with reference to reviewer comments included in decision letter(s). The following individuals involved in review of your submission have agreed to reveal their identity: Leanne M Cleaver (Reviewer #2); Jôice Dias Corrêa (Reviewer #3)

Transaction Report:

DOI: <https://doi.org/10.1128/spectrum.04327-22>

January 31, 2023

Prof. Jae-Suk Jung
Ajou University Hospital
Periodontology
164, World cup-ro
Yeongtong-gu
Suwon
Korea (South), Republic of

Re: Spectrum04327-22 (The characteristics of salivary microbiota in periodontal diseases and the potential roles of individual bacterial species to predict the severity of periodontal disease)

Dear Prof. Jae-Suk Jung:

Link Not Available

Sincerely,

Justin Kaspar

Journals Department
Reviewer comments:

Reviewer #1 (Comments for the Author):

This study investigated bacterial composition and abundances of 9 bacterial taxa of 72 individuals compared among four groups with different periodontal status. The reviewer has several concerns, especially on the study design.

1. The authors state that the microbiota differences among the four groups are derived from differences in periodontal conditions. However, significant differences were also observed in gender distribution and smoking status among the four groups. Age and the number of teeth are also different, although their p-values were slightly larger than 0.05 ($p=0.057$ and 0.07 , respectively). The reviewer could not find information on other potential confounding factors including the number of dental

caries-experienced (filled) teeth and systemic conditions such as BMI. Such confounding factors need to be strictly handled, considering the small sample size of this study.

2. The authors primarily showed the results of bacterial composition based on 16S rRNA gene sequencing analysis, followed by the qPCR result of 9 specific taxa selected based on the sequencing result. However, two characteristic taxa in the sequencing analyses, *Peptostreptococcus stomatis* and *Campylobacter showae* in Fig 2C were missed in qPCR, whereas *Parvimonas micra* was added. The authors should explain a procedure of selection of bacterial taxa for qPCR more carefully. The reviewer assumes that the primarily result of this study would be qPCR analyses of 9 specific bacterial taxa, whereas the result of 16S rRNA gene sequencing analysis only support the finding in the qPCR analyses.

3. The relative abundances in the phylum level is not informative in oral microbiota study, because each phylum includes various distinct bacteria. The reviewer recommends that the information in the phylum level remove from the manuscript, especially Abstract section.

4. Quantitative RT-PCR would indicate quantitative reverse-transcript PCR. Quantitative PCR or Quantitative real-time PCR would be better to understand, if the authors did not handle RNA.

5. General, oral and systemic conditions (age, gender, number of teeth, etc) and the results of statistical analyses should be added to Table 1.

6. The prediction performance to distinguish the severity of periodontal disease of 9 bacterial species should be examined in distinct study population to confirm the external validity.

7. Line 261, 'R. dentocarios' would be 'R. dentocariosa'.

Reviewer #2 (Comments for the Author):

Thank you for the opportunity to review your manuscript. This is overall a good piece of work, and I think that with some minor changes this will be a well-cited piece of literature. Please see my comments below:

How is this study different to the other studies that were referenced in line 71? How is this novel? What does it add?

Line 75-76 - this needs a reference - where has saliva been used for patients to self-diagnose periodontitis?

Line 118 - why was gargling used? Why not collect a whole mouth saliva sample? Is there a level of dilution here that needs to be addressed?

Line 129 - why was this database chosen and not the human oral microbiome database? Who did the 16S sequencing analysis?

Line 135 - why was 97% chosen as the threshold?

Line 225 - more diverse microbiome in severe perio? Not a dysbiosis?

Line 260 - remove overabundant

Line 261 - denticariosa - add the 'a' at the end

Line 266-308 - the chopping and changing between count and % is quite confusing and makes the results difficult to follow, especially as there is a large number of figures for this section. This would benefit from being re-written using just one of these methods.

Line 350 - you should also mention the limitations of the nucleic acid extraction method used and the bias of sequencing methods

Line 378-380 - "These results clearly indicate the gradual change of salivary microbiota to dysbiosis associated with disease progression" - but in your results you've shown an increase in diversity with disease progression?? Dysbiosis relates to increased relative abundance of genera/species, which you've shown. Therefore I think you need to reword this.

The figures need to be labelled better - axes are missing titles...

Reviewer #3 (Comments for the Author):

This study explored whether the specific bacterial species in saliva can distinguish the severity of periodontal disease by analyzing the salivary microbiota.

Introduction

1- "saliva is useful as a near-patient diagnostic tool for a point-of-care tool that allows the patient to self-diagnose the disease." The diagnosis of periodontitis involves much more than changes in the oral microbiota. Clinical information such as bleeding and probing depth and radiographic bone loss is essential. In addition to external modifying factors such as smoking, diabetes, etc. Therefore, I think it is very exaggerated to talk about the diagnosis made by the patient himself with his saliva based on microbiological changes.

Method

sequencing data together with metadata must be placed on an open access platform.

Results

the initial data with the characteristics of the sample, age, gender, must be presented in the form of a table to facilitate reading.

Discussion

The role of species such as *P. gingivalis* in periodontitis is already more than established in the literature since the classic studies by Socransky, which were not cited. Several studies with the technology of sequencing the 16s gene have already shown these associations pointed out here by the authors.

Staff Comments:

Preparing Revision Guidelines

Please return the manuscript within 60 days; if you cannot complete the modification within this time period, please contact me. If you do not wish to modify the manuscript and prefer to submit it to another journal, please notify me of your decision immediately so that the manuscript may be formally withdrawn from consideration by Microbiology Spectrum.

How is this study different to the other studies that were referenced in line 71? How is this novel? What does it add?

Line 75-76 - this needs a reference - where has saliva been used for patients to self-diagnose periodontitis?

Line 118 - why was gargling used? Why not collect a whole mouth saliva sample?

Line 129 - why was this database chosen and not the human oral microbiome database? Who did the 16S sequencing analysis?

Line 135 - why was 97% chosen as the threshold?

Line 225 - more diverse microbiome in severe perio?! Not a dysbiosis?

Line 260 - remove overabundant

Line 261 - denticariosa - add the 'a' at the end

Line 266-308 - the chopping and changing between count and % is quite confusing and makes the results difficult to follow, especially as there is a large number of figures for this section. This would benefit from being re-written using just one of these methods.

Line 350 - you should also mention the limitations of the nucleic acid extraction method used and the bias of sequencing methods

Line 378-380 - "These results clearly indicate the gradual change of salivary microbiota to dysbiosis associated with disease progression" - but in your results you've shown an increase in diversity with disease progression?? Dysbiosis relates to increased relative abundance of genera/species, which you've shown. Therefore I think you need to reword this.

The figures need to be labelled better - axes are missing titles...

The characteristics of salivary microbiota in periodontal diseases and the potential roles of individual bacterial species to predict the severity of periodontal disease

Suk Ji, Joong-Ki Kook, Soon-Nang Park, Yun Kyong Lim, Geum Hee Choi, Jae-Suk Jung

Response to Reviewer #1: This study investigated bacterial composition and abundances of 9 bacterial taxa of 72 individuals compared among four groups with different periodontal status. The reviewer has several concerns, especially on the study design.

→ We thank Reviewer #1 for your time and expert opinion.

Comment 1 The authors state that the microbiota differences among the four groups are derived from differences in periodontal conditions. However, significant differences were also observed in gender distribution and smoking status among the four groups. Age and the number of teeth are also different, although their p-values were slightly larger than 0.05 (p= 0.057 and 0.07, respectively). The reviewer could not find information on other potential confounding factors including the number of dental caries-experienced (filled) teeth and systemic conditions such as BMI. Such confounding factors need to be strictly handled, considering the small sample size of this study.

Response: Thanks for mentioning other potential confounding factors that may affect salivary microbial composition. We focused on recruiting subjects with no history of systemic disease that could influence the prognosis of periodontitis and excluded heavy smokers smoking > 10 cigarettes per day (The details about inclusion and exclusion criteria were placed at page 5, lines 91~95). Unfortunately, we thought that the confounding factors we examined would be sufficiently analyzable. As additional data, we measured the number of teeth experiencing caries by subject and presented the results in Table 1 (Page 29) with other factors. The number of teeth experiencing caries was not significantly different among groups.

Although the main findings were produced without consideration of gender

and smoking status distribution among four groups in the design, additional analyses described in the Discussion (Page 20, line 388 - Page 20 line 399) and Supplementary Tables 2 and 3 showed gender and smoking factors to have minimal or no effect on salivary microbial community profiles among groups.

However, since smoking and systemic conditions such as BMI have been considered to affect the oral microbiota (42,43), further studies should be designed to account for the possible impact of these factors on the results. These limitations were discussed on Page 21 lines 398 – 399; **However, since smoking has been considered one of the factors that affect the oral microbiota (42, 43), additional studies are needed.**

Comment 2 The authors primarily showed the results of bacterial composition based on 16S rRNA gene sequencing analysis, followed by the qPCR result of 9 specific taxa selected based on the sequencing result. However, two characteristic taxa in the sequencing analyses, *Peptostreptococcus stomatis* and *Campylobacter showae* in Fig 2C were missed in qPCR, whereas *Parvimonas micra* was added. The authors should explain a procedure of selection of bacterial taxa for qPCR more carefully.

The reviewer assumes that the primarily result of this study would be qPCR analyses of 9 specific bacterial taxa, whereas the result of 16S rRNA gene sequencing analysis only support the finding in the qPCR analyses.

Response: Among the top 11 periodontitis-related species selected based on the sequencing analyses, *Peptostreptococcus stomatis* group (<https://www.ezbiocloud.net/mtp/taxonomy?db=PKSSU4.0&tn=Peptostreptococcus%20stomatis%20group&depth=2&rg=undefined>) and *Campylobacter showae* (<https://www.ezbiocloud.net/mtp/taxonomy?db=PKSSU4.0&tn=Campylobacter%20showae%20group&depth=2&rg=undefined>) group were excluded because they are not a single species. Although *F. nucleatum* was classified into a *F. nucleatum* group in 16S rDNA amplicon sequencing, it was selected

for qPCR because of its strategic importance in biofilm formation and microbial composition. Previous studies classified *F. nucleatum* as a core species present in similar proportions in healthy people and those with periodontitis [Abusleme et al. 2021], whereas in the present study, the *F. nucleatum* group was more abundant in periodontitis patients than in healthy subjects. qPCR for *F. nucleatum* was performed to confirm the distribution according to the severity of periodontal disease.

The manuscript has been modified as follows (Page 14, line 255 ~271):

qRT-PCR was performed for periodontitis-associated *P. gingivalis*, *T. forsythia*, *T. denticola*, *P. intermedia*, *P. endodontalis*, *F. alocis*, and *F. nucleatum* and health-associated *R. dentocariosa* that showed significant differences in 16S rDNA sequencing. Among the top 11 periodontitis-related species, *Peptostreptococcus stomatis* group (<https://www.ezbiocloud.net/mtp/taxonomy?db=PKSSU4.0&tn=Peptostreptococcus%20stomatis%20group&depth=2&rg=undefined>) and *Campylobacter showae* (<https://www.ezbiocloud.net/mtp/taxonomy?db=PKSSU4.0&tn=Campylobacter%20showae%20group&depth=2&rg=undefined>) group were excluded because they are not a single species. Although *F. nucleatum* is classified as the *F. nucleatum* group in 16S rDNA sequencing, qPCR was performed because of its importance in biofilm formation and microbial structure. *P. micra* was additionally analyzed as a periodontitis-associated bacterial species, because its distribution in periodontitis decreased following non-surgical periodontal treatment in our previous study (30). Among the top 3 health-related species, only *R. dentocariosa* was analyzed by qRT-PCR because the *S. sinensis* group is not a single species

(<https://www.ezbiocloud.net/mtp/taxonomy?db=PKSSU4.0&tn=Streptococcus%20sinensis%20group&depth=2&rg=v3v4>), and *L. mirabilis* was only detected in a few samples when analyzed using a newly designed primer (data not shown).

Comment 3 The relative abundances in the phylum level is not informative in oral microbiota study, because each phylum includes various distinct bacteria. The reviewer recommends that the information in the phylum level remove from the manuscript, especially Abstract section.

Response: We agree with your opinion and have deleted the sentence about the changes in phylum level in the Abstract.

We reasoned that, even though each phylum contains a variety of distinct bacteria, gradual changes even at the phylum level with severity of periodontal disease could be analyzed as an indicator of large compositional shift in the disease-associated microbiome. These gradual changes of phylum levels according to the severity of periodontal disease can be interpreted as a characteristic of microbial dysbiosis.

Comment 4 Quantitative RT-PCR would indicate quantitative reverse-transcript PCR. Quantitative PCR or Quantitative real-time PCR would be better to understand, if the authors did not handle RNA.

Response: We have corrected quantitative RT-PCR to quantitative real-time PCR (qPCR) throughout the manuscript.

Comment 5 General, oral and systemic conditions (**age, gender, number of teeth**, etc) and the results of statistical analyses should be added to Table 1.

Response: Table 1 was modified to include general and oral characteristics (Page 29).

Comment 6 The prediction performance to distinguish the severity of periodontal disease of 9 bacterial species should be examined in distinct study population to confirm the external validity.

Response: We agree with your opinion. In order to determine the ability to distinguish the severity of periodontal disease of nine bacterial species, the prediction performance should be examined in a distinct study population to confirm the external validity. However, our study focuses on the ability of bacterial species to distinguish the severity of disease through quantitative analysis with qPCR after identifying candidate bacterial species using 16S rDNA amplicon sequencing.

The limitation of need for verification through a distinct study population to confirm external validity was additionally mentioned in the Discussion as follows (Page 18 lines 340 ~ 342); **However, the prediction performance to distinguish the severity of periodontal disease of the nine bacterial species should be investigated in a distinct large population to confirm external validity.**

Comment 7 Line 261, 'R. dentocarios' would be 'R. dentocariosa'.

Response: Line 259, 'R. dentocarios' was corrected to 'R. dentocariosa.'

Response to Reviewer #2: Thank you for the opportunity to review your manuscript. This is overall a good piece of work, and I think that with some minor changes this will be a well-cited piece of literature. Please see my comments below:

→ We thank Reviewer #2 for your time and expert opinion.

Comment	How is this study different to the other studies that were referenced in line 71?
1	How is this novel? What does it add?

Response: In references 10-12, the researchers simultaneously analyzed the distribution of periodontal bacteria in subgingival plaque and saliva and showed a positive correlation. Based on that background, our study provided evidence that the bacterial species in saliva but not subgingival plaque can distinguish the severity of periodontal disease. Moreover, this study presented a cutoff value of bacterial species to distinguish the severity of periodontal disease and its sensitivity and specificity.

Differences in the oral microbiome have been reported from closely neighboring countries [Takeshita et al. 2014]. Our study is meaningful as it shows the characteristics of the salivary microbiome according to severity of periodontal disease in a Korean population.

The importance of our study was described in the Discussion section.

Takeshita T, Matsuo K, Furuta M, Shibata Y, Fukami K, Shimazaki Y, et al. Distinct composition of the oral indigenous microbiota in South Korean and Japanese adults. Sci Rep 2014;4:6990.

Comment	Line 75-76 - this needs a reference - where has saliva been used for patients to self-diagnose periodontitis?
2	

Response: Reviewer 3 also pointed out the exaggeration of claiming self-diagnosis with saliva based on microbiological changes. We agree with the comments. The sentence was modified as follows (Page 5, lines 79 - 80); **In particular, saliva is useful as a near-patient tool for a point-of-care diagnosis (15).**

Comment Line 118 - why was gargling used? Why not collect a whole mouth saliva
3 sample? Is there a level of dilution here that needs to be addressed?

Response: The reason for collecting saliva with mouth rinse rather than unstimulated saliva was to obtain a larger amount of bacteria by washing that attached to the teeth, dorsum of the tongue, or oral mucosa through gargling. Several studies have shown that the saliva microbiome profiles are minimally affected by collection method, and mouthwash samples performed similarly to unstimulated saliva samples for analysis of the oral microbiome (Lim et al. 2017; Fan et al. 2018; Jo et al. 2019). Moreover, in some patients, saliva collection was very slow, taking more than 10 minutes. Relatively, saliva collection with mouth rinse is much faster, avoiding excessively viscous saliva [Jo et al. 2019]. For this reason, we strongly suggest that saliva collection by mouth wash should be the standard for oral microbiome analysis.

The sentence below has been added to the Material and Method section (Page 7, lines 114- 118); **Saliva was collected by the mouth rinse method to obtain more bacteria by washing away ones attached to oral surfaces, including teeth through gargling. Previous studies showed that saliva collection method had minimal effect on salivary microbial community profiles and mouth rinse samples performed similarly to unstimulated saliva samples for analysis of the oral microbiome (17-19).**

Comment 4 Line 129 - why was this database chosen and not the human oral microbiome database? Who did the 16S sequencing analysis?

Response: We obtained 16S rDNA sequencing results from the sequencing analysis company, ChunLab, Inc (<https://www.cjbioscience.com/>). They used UCHIME (20) and the non-chimeric 16S rRNA database from EzBioCloud to detect chimeric sequences on reads that contain a <97% best hit similarity rate. In EzBioCloud, 16S-based Microbiome Taxonomic Profiling (MTP) is a cloud app that allows users to generate taxonomic profiles from NGS data and to easily group and compare profiles from different samples.

NGS raw data (as FASTQ or FASTA format) were uploaded to www.ezbiocloud.net. The MTP pipeline automatically processes data that are converted to a data unit called an MTP, which represents a single metagenomic or microbiome sample. Data from public sources including the Human Microbiome Project and Short Read Archive (SRA) have been processed in advance.

Materials and Methods were modified as follows (Page 7, line 126 ~ Page 8, line 140); *Within a few days, the pellets were transferred to ChunLab, Inc (<https://www.cjbioscience.com/>) for DNA extraction and sequencing analysis. DNA was extracted using a FastDNA SPIN Kit for Soil (MPBIO), and PCR amplification was performed using primers targeting the V3 and V4 regions of the 16S rDNA. The PCR products were sequenced with an Illumina MiSeq sequencing system at ChunLab, Inc. UCHIME (20). Data from public sources including the Human Microbiome Project and Short Read Archive (SRA) have been processed in advance. Sequencing raw data (as FASTQ or FASTA format) were uploaded to www.ezbiocloud.net, and the non-chimeric 16S rRNA database from EzBioCloud was used to detect chimeric sequences on reads that contained a <97% best hit similarity rate. We generated taxonomic profiles*

from sequencing data and grouped and compared the profiles from different samples. The data for this study have been deposited in the European Nucleotide Archive (ENA) at EMBL-EBI under accession number PRJEB61123

([https://www.ebi.ac.uk/ebisearch/search?db=allebi&query=PRJEB61123%20\(ERP146221\)&requestFrom=searchBox](https://www.ebi.ac.uk/ebisearch/search?db=allebi&query=PRJEB61123%20(ERP146221)&requestFrom=searchBox)). The bacterial DNA for qPCR was sent back to the Ajou University laboratory in a storage container at 4 °C.

Comment

5

Line 135 - why was 97% chosen as the threshold?

Response:

When we asked ChunLab (<https://www.cjbioscience.com/>) for sequencing and sequence analysis, the default value for identifying identical species in sequencing analysis was a 97% threshold. To overcome this limitation of sequences with > 97% identity representing the same species, we performed qPCR for quantification of bacterial species in saliva samples.

Comment

6

Line 225 - more diverse microbiome in severe perio? Not a dysbiosis?

Response:

For clarity, the sentence was modified as follows based on the alpha diversity result (Page 12, line 223 ~225); **When alpha diversity metrics were applied using the number of identified species, Chao 1, Shannon index, and phylogenetic diversity, the diversity tended to increase as the severity of periodontal disease increased (Figure 1).**

The 16S rDNA sequencing revealed that the microbial composition of saliva tends toward dysbiosis with the transition from health to severe periodontitis. Analysis of the relative abundance at the species level showed that

periodontitis-associated bacterial species were significantly more abundant as the severity of periodontal disease increased, while health-associated species were significantly more abundant in the healthy group. These results clearly indicate the gradual change of salivary microbiota to dysbiosis with disease progression.

Comment

7

Line 260 - remove overabundant

Response:

Yes, “overabundant” was removed from the sentence you pointed.

Comment

8

Line 261 - denticariosa - add the 'a' at the end

Response:

Page 14, Line 267 – ‘denticarios’ was corrected to ‘dentocariosa’

Comment

9

Line 266-308 - the chopping and changing between count and % is quite confusing and makes the results difficult to follow, especially as there is a large number of figures for this section. This would benefit from being re-written using just one of these methods.

Response:

We are sorry for the confusion about qPCR results. As the severity of periodontal disease increased, the total number of bacteria in saliva increased, resulting in an increase of all bacterial species, including health-associated ones, while the relative abundance of periodontitis-associated or health-associated bacterial species respectively increased or decreased with increasing severity of periodontal disease. Therefore, the count and relative abundance of a specific bacterial species have different meanings in terms of disease

progression and the ability to distinguish the severity of disease.

The pattern of change in bacterial count according to severity of periodontal disease was different for each bacterial species, and this may be a unique characteristic by bacterial species according to the severity of periodontal disease. For example, *P. endodontalis* and *P. intermedia* showed relatively high AUC, sensitivity, and specificity for distinguishing D1 and D3, respectively. In prediction performance analyzed by ROC curve, the nine-count of *P. endodontalis* in D1 showed 0.82, 0.64, and 1.0 values of AUC, sensitivity, and specificity, respectively. This means that the presence of a less than nine-count of *P. endodontalis* is a healthy condition. Also, *P. intermedia* can be a good indicator of severe periodontitis, as the 18 count in D3 showed 0.8 AUC, 0.86 sensitivity, and 0.74 specificity. Figure 4B shows that the distribution of *P. intermedia* only changed in the SP group.

The increase or decrease of relative abundance of a specific bacterial species can show the compositional changes of salivary microbiome during dysbiosis. In this respect, the relative abundance of bacterial species can be more accurate to distinguish the severity of periodontal disease than the bacterial count. For this reason, we analyzed and presented both values (qPCR (count) and relative abundance (qPCR (%))) in this manuscript.

However, the paragraph has been rewritten for clarity (Page 14, line 272 ~ Page 16, line 305); **Bacterial count was calculated as the number of bacteria in 11.43 µl of mouth rinsing saliva sample, considering that 2 µl of the DNA template was used in qPCR, and the relative abundance of bacterial species was calculated as the ratio of specific bacterial count to total bacterial count. The correlation between relative abundance (%) by 16S rDNA sequencing and that by qPCR (%) was analyzed for each bacterial species. A significant positive correlation was found in all bacterial species (Figure 3). However, there were some differences in the correlation coefficient R²-value; the R²-**

values of *P. gingivalis*, *T. forsythia*, *P. intermedia*, and *F. alocis* were greater than 0.7, showing relatively strong correlations. *F. nucleatum* showed the lowest R^2 -value, probably because the group was analyzed using 16S rDNA sequencing that includes several species of *F. nucleatum*, *F. polymorphum*, *F. vincentii*, *F. animalis*, *F. simiae*, *F. canifelinum*, and *F. hwasookii* (<https://www.ezbiocloud.net/mtp/taxonomy?db=PKSSU4.0&tn=Fusobacterium%20nucleatum%20group&rg=V3V4>). Based on the genome-based approach, *F. nucleatum* subsp. *nucleatum*, subsp. *polymorphum*, subsp. *vincentii*, and subsp. *animalis* were classified respectively as *F. nucleatum*, *F. polymorphum*, *F. vincentii*, and *F. animalis* (31). qPCR (%) of *P. gingivalis*, *T. forsythia*, *P. intermedia*, and *F. alocis* showed significant differences among groups ($p < 0.01$). Simple linear regression analysis showed that qPCR (%) of *R. dentocariosa* decreased as the severity of disease increased ($p < 0.01$) (Figure 4A). In qPCR (count), all bacterial species except *R. dentocariosa* were significantly different among groups and increased as the severity of disease increased (Figure 4B).

Next, we analyzed whether the qPCR (%) or qPCR (count) of bacterial species was correlated with the sum of PD. The sum of PD significantly increased with severity of disease (Table 1). The qPCR (%) or qPCR (count) of all bacterial species except *R. dentocariosa* showed a tendency of positive correlation with sum of PD. In common, *P. gingivalis*, *T. forsythia*, and *F. alocis* showed relatively high R^2 -values greater than 0.3 (Figure 5).

The prediction performance to distinguish the severity of periodontal

disease of nine bacterial species was evaluated using an ROC curve. *P. gingivalis* showed an AUC of 0.73 to 0.82 and a sensitivity and specificity greater than 72% in all divisions (Table 2). In D1, the qPCR (%) and qPCR (count) of *F. alocis* and *P. endodontalis* showed AUC values larger than 0.8. Especially, 0.004% of *F. alocis* distinguished the healthy group with a sensitivity of 0.84 and specificity of 0.88, and five of the qPCR (count) of *F. alocis* distinguished the healthy group with a sensitivity of 0.89 and specificity of 0.88. In D3, the qPCR (count) of *T. forsythia* and *P. intermedia* showed relatively high AUC values larger than 0.8. A qPCR (count) of 213 for *T. forsythia* distinguished the SP with a sensitivity of 0.62 and specificity of 0.98 (Table 2).

Comment Line 350 - you should also mention the limitations of the nucleic acid
10 extraction method used and the bias of sequencing methods

Response: The sentence has been modified as “These differences among studies may be caused by ethnic or geographic differences in subjects, saliva sampling methods, nucleic acid extraction methods, bias of sequencing methods, diet, bacterial detection methods, local genetic differences in bacteria, or pathogenicity of bacterial strains.” (Page 18, line 335 ~338)

Comment Line 378-380 - "These results clearly indicate the gradual change of salivary
11 microbiota to dysbiosis associated with disease progression" - but in your results you've shown an increase in diversity with disease progression?? Dysbiosis relates to increased relative abundance of genera/species, which you've shown. Therefore I think you need to reword this.

Response: With an increase in alpha diversity, gradual changes at the genera and species levels were observed as the severity of periodontal disease increased. In that paragraph, we wanted to discuss the gradual change in phylum level. However, we decided that it would be better to delete the noted sentence.

Comment
12 The figures need to be labelled better - axes are missing titles...

Response: The axis titles were added in Figure 2 and Supplementary Tables 2 and 3.

Response to Reviewer #3: This study explored whether the specific bacterial species in saliva can distinguish the severity of periodontal disease by analyzing the salivary microbiota.

Comment
1 Introduction
1- "saliva is useful as a near-patient diagnostic tool for a point-of-care tool that allows the patient to self-diagnose the disease."

The diagnosis of periodontitis involves much more than changes in the oral microbiota. Clinical information such as bleeding and probing depth and radiographic bone loss is essential. In addition to external modifying factors such as smoking, diabetes, etc. Therefore, I think it is very exaggerated to talk about the diagnosis made by the patient himself with his saliva based on microbiological changes.

Response: We agree with you about the exaggeration of claiming self-diagnosis with saliva based on microbiological changes. We cautiously predicted that periodontal disease accompanied by change of microbial composition could be self-diagnosed using salivary microbiological biomarkers, as is achieved with COVID-19.

As you pointed out, that sentence has been modified as follows (Page 5, lines 79 - 80); **In particular, saliva is useful as a near-patient tool for a point-of-care diagnosis (15).**

Comment Method

2

sequencing data together with metadata must be placed on an open access platform.

Response: Page 8 lines 136 ~ 139; **The data for this study have been deposited in the European Neucleotide Archive (ENA) at EMBL-EBI under accession number PRJEB61123 ([https://www.ebi.ac.uk/ebisearch/search?db=allebi&query=PRJEB61123%20\(ERP146221\)&requestFrom=searchBox](https://www.ebi.ac.uk/ebisearch/search?db=allebi&query=PRJEB61123%20(ERP146221)&requestFrom=searchBox)).**

Comment Results

3

the initial data with the characteristics of the sample, age, gender, must be presented in the form of a table to facilitate reading.

Response: As you pointed out, the initial data were presented in Table 1.

Comment Discussion

4

The role of species such as *P. gingivalis* in periodontitis is already more than established in the literature since the classic studies by Socransky, which were not cited. Several studies with the technology of sequencing the 16s gene have already shown these associations pointed out here by the authors.

Response: As you pointed out, several studies including sequencing analysis have shown the association of the nine bacterial species of interest in this study with periodontal disease. Although this study did not find any new species, it suggested the relative importance of bacterial species in its association with periodontal disease or the ability to distinguish the severity of periodontal disease. Among the nine bacterial species of interest, *P. gingivalis*, *T. forsythia*, and *F. alocis* were superior for diagnosis of disease. *T. denticola*, a red complex bacterial species, did not show a strong association with disease. This was an unexpected finding because the association of red-complex bacteria in subgingival plaque with periodontitis has been well identified.

The importance of this study was described in the Discussion section.

April 27, 2023

Prof. Jae-Suk Jung
Ajou University Hospital
Periodontology
164, World cup-ro
Yeongtong-gu
Suwon
Korea (South), Republic of

Re: Spectrum04327-22R1 (The characteristics of salivary microbiota in periodontal diseases and the potential roles of individual bacterial species to predict the severity of periodontal disease)

Dear Prof. Jae-Suk Jung:

Your manuscript has been accepted, and I am forwarding it to the ASM Journals Department for publication. You will be notified when your proofs are ready to be viewed.

Sincerely,

Justin Kaspar
Editor, Microbiology Spectrum
